# [18F]FDG PET/CT Integration in Evaluating Immunotherapy for Lung Cancer: A Clinician’s Practical Approach

**DOI:** 10.3390/diagnostics14182104

**Published:** 2024-09-23

**Authors:** Juliette Brezun, Nicolas Aide, Evelyne Peroux, Jean-Laurent Lamboley, Fabrice Gutman, David Lussato, Carole Helissey

**Affiliations:** 1Department of Medical Oncology and Clinical Research Unit, Military Hospital Bégin, 94160 Saint-Mandé, France; 2INSERM ANTICIPE U1086, Caen University, 14000 Caen, France; nicolasaide0@gmail.com; 3Department of Radiology, Military Hospital Laveran, 13013 Marseille, France; evelyne.peroux@intradef.gouv.fr; 4Department of Radiology, Military Hospital Bégin, 94160 Saint-Mandé, France; jean-laurent.lamboley@intradef.gouv.fr; 5Department of Nuclear Medicine, Paul d’Egine Hospital, 94500 Champigny-sur-Marne, France; fgutman@cmn-idf.fr; 6Department of Nuclear Medicine, Centre Cardiologique du Nord, 93200 Saint-Denis, France; davidlussato@gmail.com

**Keywords:** immune checkpoint inhibitors, lung cancer, treatment response assessment, [18F]FDG PET/CT imaging

## Abstract

The advent of immune checkpoint inhibitors (ICIs) has revolutionized the treatment paradigm of lung cancer, resulting in notable enhancements in patient survival. Nevertheless, evaluating treatment response in patients undergoing immunotherapy poses distinct challenges due to unconventional response patterns like pseudoprogressive disease (PPD), dissociated response (DR), and hyperprogressive disease (HPD). Conventional response criteria such as the RECIST 1.1 may not adequately address these complexities. To tackle this issue, novel response criteria such as the iRECIST and imRECIST have been proposed, enabling a more comprehensive assessment of treatment response by incorporating additional scans and considering the best overall response even after radiologic progressive disease evaluation. Additionally, [18F]FDG PET/CT imaging has emerged as a valuable modality for evaluating treatment response, with various metabolic response criteria such as the PERCIMT, imPERCIST, and iPERCIST developed to overcome the limitations of traditional criteria, particularly in detecting pseudoprogression. A multidisciplinary approach involving oncologists, radiologists, and nuclear medicine specialists is crucial for effectively navigating these complexities and enhancing patient outcomes in the era of immunotherapy for lung cancer. In this review, we delineate the key components of these guidelines, summarizing essential aspects for radiologists and nuclear medicine physicians. Furthermore, we provide insights into how imaging can guide the management of individual lung cancer patients in real-world multidisciplinary settings.

## 1. Introduction

Lung cancer is recognized as the leading cause of cancer-related deaths, accounting for 1.8 million deaths (18.7%), with 2 480 675 new cases diagnosed in 2022 [1]. Similar to many tumor models, the treatment landscape of lung cancer has been revolutionized by the development of immune checkpoint inhibitors (ICIs), initially as monotherapy and later in combination therapy, which has significantly improved patient survival, first in the metastatic setting and now in the localized stage. With the implementation of immunotherapy, the five-year survival rate is observed at 29%, representing a 2% increase compared to previous data [2]. 

ICIs disrupt immune regulation mechanisms that traditionally inhibit antitumor immune responses. They primarily act by blocking interactions between surface receptors on T lymphocytes, such as PD-1 (programmed cell death protein 1) and CTLA-4 (cytotoxic T-lymphocyte-associated protein 4), and their ligands, including PD-L1 (programmed death ligand 1). Through this mechanism involving the tumor microenvironment, cancer assessment has been profoundly disrupted. 

The utilization of ICIs in lung cancer patients presents a significant challenge for the medical imaging community in assessing treatment response. In fact, clinicians are thus confronted with various dilemmas, such as whether to discontinue treatment prematurely or to subject a patient to potential side effects for too long when disease control is not achieved. This endeavor requires a collaborative effort involving oncologists, radiologists, and nuclear medicine specialists. Emerging evidence suggests that [18F]FDG PET/CT is a valuable tool for evaluating treatment response in this context. Recently published joint international guidelines provide recommendations for the use of [18F]FDG PET/CT imaging during immunomodulatory treatments in patients with solid tumors. In this review, we outline the key elements of these guidelines, summarizing essential aspects for radiologists and nuclear medicine physicians. Additionally, we offer insights into how this technology can inform the management of individual lung cancer patients in real-world multidisciplinary settings.

## 2. RECIST 1.1 and iRECIST for Assessing Immunotherapy in Lung Cancer

The purpose of the RECIST 1.1 is to assess tumor response to anti-cancer treatments by providing standardized criteria for measuring changes in tumor lesion size. These criteria enable the determination of complete response, partial response, stable disease, or progressive disease based on the evolution of tumor size compared to the initially measured size. They provide a reliable, simple, and reproducible standardized method for evaluating tumor response. The objective is to facilitate a comparison of imaging studies and enable an objective assessment of treatment efficacy [3].

Target lesions are chosen and identified during CT or MRI examination before the start of treatment (baseline) and are used throughout the follow-up, with a maximum of five lesions per patient and two lesions per organ. These lesions must be easily recognizable and measurable in the acquisition plane, with a greater diameter of at least 10 mm and, for lymph nodes, a smaller diameter of at least 15 mm. The sum of the longest diameters (SLD) of all target lesions is calculated and serves as a reference during follow-up.

Non-target lesions correspond to lesions that do not meet the criteria for defining target lesions according to RECIST guidelines, including small lesions and non-measurable lesions. Although they are not directly used to assess treatment response, monitoring and evaluating them are important for an overall assessment of disease and treatment response in cancer patients.

The different types of tumor response may include the following:Complete response (CR): disappearance of all target and non-target lesions, with no lymph node measuring more than 10 mm in smaller diameter.Partial response (PR): significant reduction (at least 30%) in the SLD compared to baseline, without unequivocal progression of non-target lesions.Stable disease (SD): no significant progression of target or non-target lesions.Progressive disease (PD): significant increase (at least 20%) in the SLD and unequivocal progression of non-target lesions or appearance of new lesions compared to the examination with the smallest sum since the start of treatment (nadir).

The assessment using the RECIST 1.1 is highly valuable for evaluating anti-cancer treatments such as chemotherapy or targeted therapy (Figure 1).

Topalian et al. reported the first case of initial progression under anti-PD-1 antibody treatment, which would have led to treatment discontinuation after 2 months. However, continuing the treatment unchanged resulted in a near-complete response at 4 months [4].

This case introduced new concepts. ICIs have been associated with unconventional response patterns such as pseudoprogressive disease (PPD), dissociated response (DR), and hyperprogressive disease (HPD). Additionally, a distinct characteristic of ICIs compared to conventional chemotherapy and/or molecularly targeted therapies is the occurrence of durable response.

### 2.1. Pseudoprogressive Disease (PPD)

This response pattern, termed pseudoprogressive disease (PPD), prompted the development of new response criteria. PPD typically occurs most frequently within the initial 4–6 weeks of treatment but may also manifest several months after ICI initiation. 

The incidence of PPD varies among tumor types and immunotherapies, with reports of up to 10% of patients based on CT scan or [18F]FDG PET (Ref). In the field of NSCLC, radiological PPD has been reported to occur rarely: in a retrospective, multicentric study having been conducted in a real-word setting and involving 542 patients, Fujimoto et al. observed radiological PPD in 14 of 542 patients (3%) treated with anti-programmed death 1 (PD-1)/programmed death ligand 1 (PD-L1) agents [5].

This reassuring clinical situation with a relatively good prognosis contrasts with the concept of disease hyperprogression, suggesting potentially harmful effects of these drugs [6].

### 2.2. Hyperprogressive Disease (HPD)

HPD is defined by accelerated tumor growth kinetics, leading to premature death. The prevalence of HPD varies, ranging from 1 to 30%, with some risk factors identified, including higher age and specific genetic aberrations [6,7,8,9,10]. 

Thus, the RECIST 1.1 appeared to be inadequate for the evaluation of immunotherapy.

In response to this issue, Seymour et al. introduced the iRECIST, in which the Guidelines to Assess the Clinical Benefit of Cancer Immunotherapy are refined. The authors underscore the unique challenges associated with assessing response to immunotherapies due to their unconventional response patterns, such as pseudoprogression [11]. The iRECIST guidelines aim to standardize this assessment by addressing these unusual response patterns. They establish specific criteria for evaluating tumor response, including defining progression, response, and disease stabilization criteria, as well as providing guidelines for managing cases of pseudoprogression. These guidelines are intended to enhance the consistency and accuracy of evaluating response to immunotherapies in clinical trials, potentially impacting the management of cancer patients and drug regulation in this rapidly evolving field.

Thus, immunotherapy, aimed at stimulating the anti-tumor immune response, can lead to an initial increase in lesion size or even the appearance of new lesions, followed by a subsequent decrease in these lesions, thus introducing the concept of pseudoprogression. As these pseudoprogressions can only be diagnosed retrospectively due to the lack of radiological and clinical specificity, new imaging evaluation criteria have been proposed and termed the iRECIST. These criteria apply when RECIST progression is observed and four new response categories are defined:**iUPD (immune unconfirmed progressive disease)**: A determination of new target and non-target lesions according to the RECIST takes place if necessary. The patient remains in progression compared to nadir but does not meet the aforementioned progression criteria. If clinical oncological evaluation allows, treatment is continued, and a new imaging assessment is performed between 4 and 8 weeks, after which the following response categories may be observed:**iCPD (immune confirmed progressive disease)**: If progression of lesions (target, non-target, or new lesions) defined in iUPD is observed, it is compared to the iUPD scan according to the following criteria:○An additional increase in the sum of target lesions and/or new target lesions greater than or equal to 5 mm;○An additional qualitative increase in non-target lesions and/or new non-target lesions;○Appearance of new lesions;○Additional progression according to the RECIST on a different type of lesion than those defined in iUPD.**iCR (immune complete response):** disappearance of all target, non-target, and new lesions, with all lymph nodes measuring less than 10 mm in short axis.**iPR (immune partial response):** decrease of more than 30% in the sum of target lesions compared to pre-treatment, and non-target and new lesions do not meet the criteria for progression or no longer meet them.**iSD (immune stable disease):** target lesions meet the RECIST for stable disease compared to pre-treatment (decrease of less than 30%) and nadir (increase of no more than 20%), and non-target and new lesions do not meet progression criteria.

In case of response, any new progression becomes iUPD again and must be confirmed: a patient can only be defined as having iCPD if they had iUPD previously.

Figure 2 summarizes the therapeutic strategy according to the iRECIST.

To go a step further and refine the radiographic evaluations and their impact on survival, Hodi et al. defined the immune-modified Response Evaluation Criteria In Solid Tumors (imRECIST) [12]. In summary, the imRECIST allow for the inclusion of additional scans and considers best overall response (BOR) even after radiologic progressive disease (PD) assessment in patients undergoing continuous treatment. New lesions (NLs) are counted alongside target lesions (TLs) when measurable, and their presence does not necessarily signify PD if the TLs are stable. Progression in non-target lesions (NTLs) does not indicate PD. In imRECIST-defined progression-free survival (imPFS) analysis, PD or death is considered an event; however, if a subsequent scan shows stable disease (SD), partial response (PR), or complete response (CR) at least four weeks later, an initial imRECIST PD is not considered an imPFS event. An imRECIST PD without further assessments is deemed an imPFS event [12].

Prolongation of progression-free survival (PFS) as per the imRECIST compared to RECIST v1.1 was linked to either longer or comparable overall survival (OS). Patients exhibiting a spike pattern, characterized by an initial increase followed by a decrease in target lesions, demonstrated longer OS compared to those without target lesion reversion [12].

Table 1 summarizes the RECIST 1.1, iRECIST, and imRECIST.

The situation of pseudoprogression remains challenging throughout the treatment process, not just at the initial assessment.

While these criteria consider pseudoprogression, they do not fully capture the complexity of managing patients on ICIs, as several emerging patterns of response and progression are discussed below. Furthermore, a diverse array of treatment combinations is presently under investigation. 

This raises the following question: can FDG PET assist clinicians in patient management?

## 3. Role of PET/CT in the Management of Lung Cancer Patients Receiving Immunotherapy

The interpretation of [18F]FDG PET/CT in patients undergoing immunotherapy must consider the phenomenon of pseudoprogressive disease, as defined in a CT scan (PPD or iUPD). 

Therefore, several metabolic response criteria have been proposed as alternatives to the PERCIST, which are typically recommended for evaluating chemotherapy and molecularly targeted therapies. These criteria are outlined above. Currently, there are insufficient data to determine the preferable approach for categorizing response, and the impact on long-term patient outcomes has not been prospectively validated in randomized clinical trials. Some alternatives to the PERCIST are outlined below:**PERCIMPT:** To overcome the limitations of [18F]FDG PET imaging, particularly the issue of pseudoprogression in assessing immunotherapy response in melanoma, the PET Response Evaluation Criteria for Immunotherapy (PERCIMPT) were developed. The PERCIMPT highlight that changes in the absolute number of [18F]FDG-avid lesions are more indicative of clinical outcomes than alterations in standardized uptake values (SUVs) during immunotherapy. According to these criteria, increases in SUVs or the appearance of a single new hypermetabolic lesion in follow-up [18F]FDG PET/CT scans does not necessarily signal disease progression, as suggested by the conventional PERCIST/EORTC criteria. Instead, the presence of four newly developed [18F]FDG-avid lesions, with the threshold for the number of lesions decreasing as their functional diameter increases, can more accurately identify patients with progressive disease.**imPERCIST:** Based on the PERCIST, the imPERCIST differ in that the emergence of new lesions alone is not sufficient to classify a patient as having progressive disease. These criteria require the calculation of the sum of peak standardized uptake values normalized for lean body mass (SULpeak) for up to five lesions in baseline and follow-up scans (with a maximum of two per organ). Target lesions in follow-up scans are the most intense lesions and may differ from those identified at baseline. Progressive metabolic disease (PMD) is defined as an increase of at least 30% in the sum of SULpeak. The appearance of new lesions alone does not define PMD; instead, new lesions are included in the SULpeak only if they show higher uptake than existing target lesions or if fewer than five target lesions were present in the baseline scan.**iPERCIST:** Recently, some studies have adapted the PERCIST to incorporate the “wait-and-see” approach initially proposed by the iRECIST guidelines, resulting in the iPERCIST. Under these criteria, patients showing new lesions or a greater than 30% increase in the SULpeak or in the SULpeak of the most intense lesions are classified as having unconfirmed progressive metabolic disease (uPMD). A subsequent reassessment 4 to 8 weeks later is required to confirm progressive metabolic disease (cPMD). Studies indicate that for patients with metastatic lung cancer who exhibit uPMD in the initial interim PET, a confirmatory PET reclassifies about one-third of these early-progressing patients as having atypical response patterns (PPD or dissociated response), who ultimately benefit from continued ICI therapy. This highlights the danger of prematurely concluding treatment failure after an initial uPMD, a risk that is particularly pronounced with [18F]FDG PET/CT due to its heightened sensitivity in detecting immune cell activation.

As a general recommendation from the joint EANM/ANZNM/SNMMI guideline/procedure standard [13], in cases where doubts exist between progression or pseudoprogression, especially during the first post-treatment evaluation, a confirmatory follow-up [18F]FDG PET/CT study 4–8 weeks later in a clinically stable setting should be performed (Figure 3 and Figure 4). This consensus arises from the lack of a robust and externally validated tool to differentiate true progression from pseudoprogression based on a single imaging assessment. Therefore, treatment should be maintained in clinically stable patients, unless excessive toxicity occurs, to avoid prematurely discontinuing ICIs in patients who might eventually experience clinical improvement and an objective response.

Table 2 summarizes the PERCIST 1.0, PECRIT, PERCIMT, iPERCIST, and imPERCIST5.

## 4. In Practice

### 4.1. The Role of PET in the Care Pathway

The use of [18F]FDG PET/CT within immunotherapeutic protocols should be carefully considered at various points in line with treatment, guided by clinical needs (Figure 5).

Prior to treatment initiation, [18F]FDG PET/CT should be regarded as essential for tumor evaluation, especially for [18F]FDG-avid tumors, particularly in cases of first-line immunotherapeutic regimens. This assessment serves as a foundation for monitoring tumor status or confirming disease progression/recurrence. It is recommended to define target lesions, to define the areas of highest [18F]FDG uptake (e.g., SUVmax and SUVpeak), and to compute volumetric parameters (e.g., MTV) at baseline, providing a basis for monitoring disease response over time but also a prognostic factor. In addition to PET response criteria tailored or adjusted to addressing the challenges posed by immunotherapy, various research groups have identified baseline prognostic factors for response, such as SUV metrics and indicators of immune activation.

Interestingly, patients with NSCLC presenting with a higher baseline SUVmax may have an increased likelihood of responding to treatment with ICIs. The association between metabolic variables and immune cell expression in the tumor microenvironment, as well as its correlation with progression-free survival, was initially reported by Rossi et al. [21] and Grizzi et al. [22] and subsequently confirmed by Kaira et al. [23], suggesting that SUV metrics from 18F-FDG PET/CT scans could serve as potential predictors for selecting candidates for immunotherapy.

Regarding signs of immune activity, the primary indicator to consider is spleen enlargement and/or increased uptake leading to an inversion of the spleen-to-liver ratio (SLR) [14,24,25,26]. Some groups have also proposed additional indicators such as the bone marrow-to-liver ratio (BLR) and uptake in the ileocecal valve [27,28]. Despite appearing counterintuitive (with increased spleen uptake often interpreted as indicative of “unleashed” T lymphocytes and thus a better outcome), two studies [25,29] have shown that an increase in the SLR is actually detrimental, with patients exhibiting this pattern experiencing worse survival. This elevated SLR observed at baseline or early in the course of ICI treatment is believed to be associated with inflammation.

Another valuable piece of information gleaned from PET imaging is the occurrence of immune-related adverse events (irAEs). While several studies have indicated that patients experiencing irAEs may have better survival rates [30,31], research on PET-detected irAEs is limited. Recently, Iravani et al. [32] demonstrated in melanoma patients receiving a combination of two ICIs that 18F-FDG PET/CT scans often revealed relevant irAEs, which sometimes preceded clinical diagnosis. Special attention should be paid to irAEs requiring treatment withdrawal or immunosuppressive treatment, i.e., colitis and pneumonitis [33].

During the treatment course, interim [18F]FDG PET/CT scans are advised, typically scheduled 8–12 weeks (i.e., 3–4 cycles) after treatment initiation [13]. This is particularly valuable for complementing information obtained from morphological imaging with CT and resolving discordant findings. Differentiating between disease progression and pseudoprogression necessitates a follow-up scan 4–8 weeks later under conditions of clinical stability, underscoring the importance of transparent communication with the treating clinician. Alternatively, a biopsy of the radiographically/metabolically progressive lesion may be warranted. PET/CT scans may also be conducted earlier or later during treatment in instances of clinical deterioration or suspected progression identified in contrast-enhanced CT. It can also guide in the management of dissociated response (DR). DR involves a decrease or stabilization in some tumor sites alongside an increase in others. Although DR is less common, it has been reported in up to 10% of cases treated with ICIs. From a clinical perspective, patients with dissociated response may benefit from treatment beyond progression potentially by continuing checkpoint inhibitor therapy and integrating local treatments, such as surgery, radiotherapy, or interventional radiological treatment of oligoprogressive lesions.Before and after the discontinuation of immunotherapy, in patients receiving maintenance therapy or undergoing long-term treatment with ICIs, obtaining [18F]FDG PET/CT scans may help assess metabolic response, particularly in cases of partial responders or stable disease in CT.

In patients requiring a temporary interruption of immunotherapy, for instance in the case of irAEs requiring withdrawal and/or corticosteroid or immunosuppressive treatment, [18F]FDG PET/CT restaging is recommended before restarting the treatment to re-establish a new baseline for subsequent response assessment. [18F]FDG PET/CT can be used to check the complete resolution of severe irAEs (pneumonitis and colitis).

### 4.2. Insights from a Multidisciplinary Staff Meeting Real-Life Experience: Navigating in the Maze of PPD

An adequate awareness on how to utilize [18F]FDG PET/CT should be part of the basic knowledge base of oncologists involved in delivering immunotherapy and is vital for cancer imaging specialists. As with many other clinical indications in nuclear medicine, a multidisciplinary approach is important to provide clinical context when imaging findings raise the possibility of PPD or HPD or irAEs are suspected. In the latter case, open communication channels with the managing clinician are critical to optimally manage unexpected events.

Figure 6 summarizes a proposed care pathway for patients with lung cancer receiving immunotherapy.

## 5. Conclusions

ICIs have revolutionized the treatment landscape of lung cancer, significantly improving patient survival rates. However, the unconventional response patterns associated with ICIs, such as pseudoprogressive disease (PPD), dissociated response (DR), and hyperprogressive disease (HPD), pose significant challenges in assessing treatment efficacy using traditional response criteria like the RECIST 1.1.

To address these challenges, new response criteria have been proposed, such as the iRECIST and imRECIST, which aim to standardize the assessment of tumor response to immunotherapies. These criteria consider additional scans and allow for the best overall response even after radiologic progressive disease assessment, providing a more comprehensive evaluation of treatment response.

Furthermore, [18F]FDG PET/CT imaging has emerged as a valuable tool for evaluating treatment response in lung cancer patients undergoing immunotherapy. Various metabolic response criteria, such as the PERCIMT, imPERCIST, and iPERCIST, have been developed to address the limitations of traditional response criteria, particularly in detecting pseudoprogression.

In clinical practice, [18F]FDG PET/CT scans play a crucial role at various stages of the patient’s treatment journey, from baseline evaluation to monitoring response during and after treatment. These scans not only aid in distinguishing between true progression and pseudoprogression but also provide valuable prognostic information and help detect immune-related adverse events.

Overall, a multidisciplinary approach involving oncologists, radiologists, and nuclear medicine specialists is essential for effectively navigating the complexities of assessing treatment response in lung cancer patients receiving immunotherapy. By incorporating advanced imaging techniques like [18F]FDG PET/CT and adopting standardized response criteria, clinicians can better tailor treatment strategies and improve patient outcomes in this rapidly evolving field.

## Figures and Tables

**Figure 1 diagnostics-14-02104-f001:**
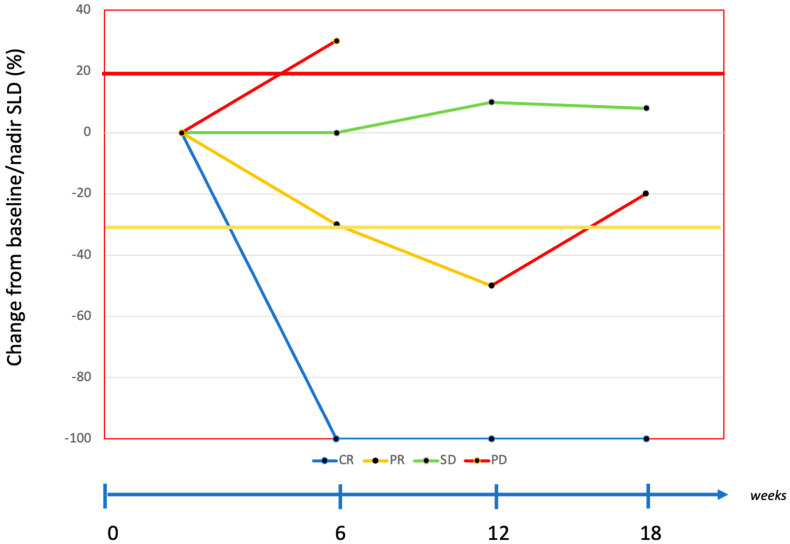
Tumor response according RECIST 1.1. SLD: sum of the longest diameters. CR: complete response. PR: partial response. SD: stable disease. PD: progressive disease.

**Figure 2 diagnostics-14-02104-f002:**
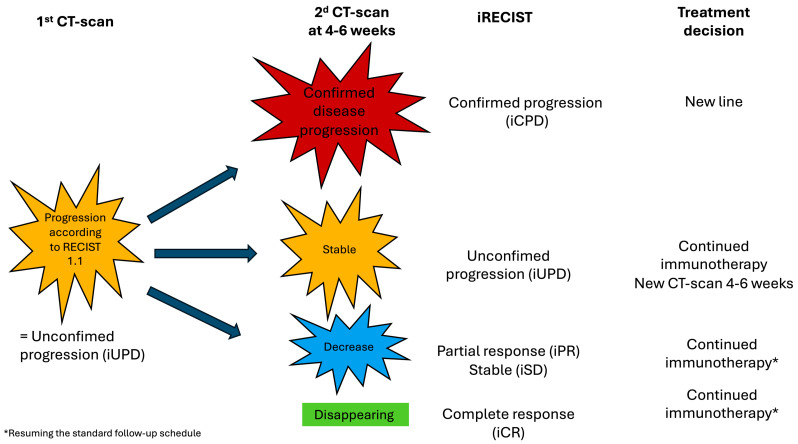
iRECIST.

**Figure 3 diagnostics-14-02104-f003:**

**PET/CT imaging of patient with metastatic non-small cell lung cancer: evaluating pseudoprogression during treatment**. Whole-body maximum intensity projection views are presented in panels (**a**,**c**,**e**). Corresponding fused transaxial PET/CT slices at the level of the thoracic disease and the left adrenal metastasis are shown in panels (**b**,**d**,**f**). (**a**,**b**) Baseline PET/CT images of a 58-year-old male patient with metastatic non-small cell lung cancer, showing a left upper lobe tumor (head arrow), bulky nodal disease (red arrow), and oligometastatic disease with a soft tissue lesion adjacent to the right hip and left adrenal metastasis (red dotted arrow). (**c**,**d**) Two months after initiation of chemotherapy plus immunotherapy, partial metabolic response is observed, with increased tracer uptake in the left adrenal metastasis (red dotted arrow). According to conventional criteria (PERCIST or EORTC), this would classify the patient as having progressive disease. However, using imPERCIST, the patient is classified as a partial metabolic responder. As the patient did not experience deterioration in performance status, treatment was continued. Evaluation two months later (**e**,**f**) revealed a complete metabolic response of distant metastases and nodal disease, along with an almost complete metabolic response of the primary tumor (head arrow).

**Figure 4 diagnostics-14-02104-f004:**
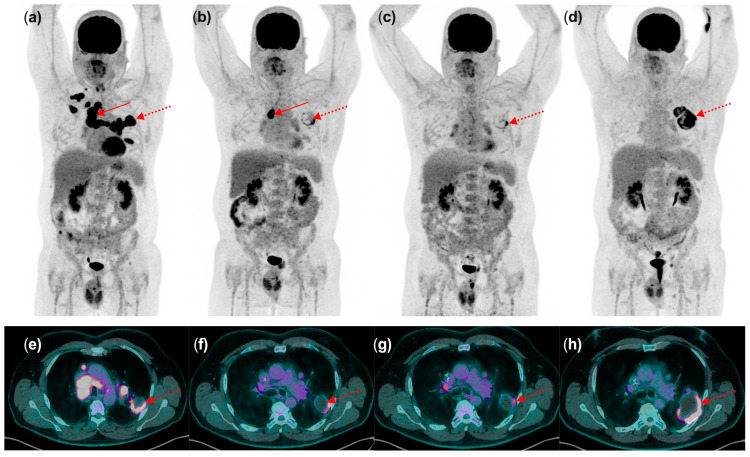
The usefulness of PET/CT imaging of patient with metastatic non-small cell lung cancer: patient experiencing good response to treatment followed by progression. A 54-year-old male patient diagnosed with metastatic adenocarcinoma of the left upper lobe treated with chemotherapy (paclitaxel and carboplatin) and immunotherapy (pembrolizumab). 18F-FDG PET/CT (**a**–**d**) maximum-intensity views depicting at baseline (**a**) high uptake with a rim-like pattern in the primary tumor (red dotted arrow), along with bulky nodal disease involving the mediastinum and the left hilum (red arrows), as well as two bone metastases in the axial and appendicular skeleton. 18F-FDG PET/CT after 3 cycles of treatment (**b**) shows a partial metabolic response of the primary lesion (red dotted arrow) and the nodal disease with only one residual nodal uptake remaining at the level of station 2R (red arrow). Additionally, complete metabolic response is noted in both bone lesions. Subsequent 18F-FDG PET/CT after 5 months of treatment (**c**) shows only residual uptake in the primary tumor (red dotted arrow). Maintenance immunotherapy was initiated, and a follow-up PET examination at 10 months reveals an increase in both the size and metabolic activity of the primary tumor (red dotted arrow), with adjacent rib involvement (**d**). Corresponding fused PET/CT transverse slices at the level of the primary tumor (red dotted arrow) are depicted in panels (**e**–**h**).

**Figure 5 diagnostics-14-02104-f005:**
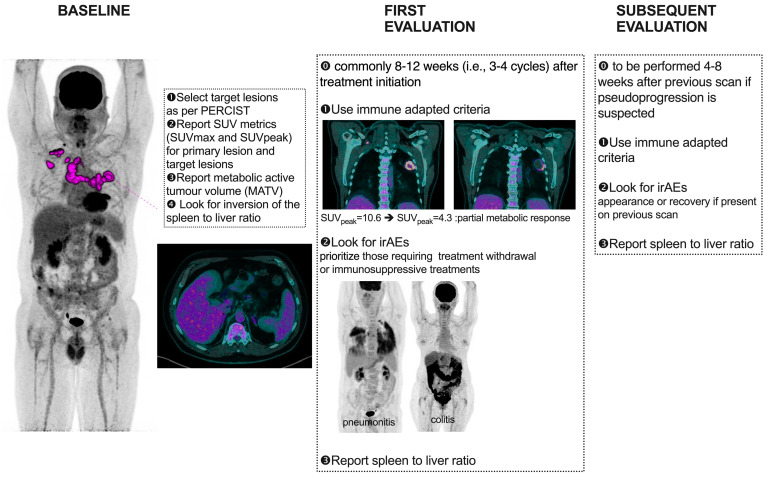
A checklist for the PET reader.

**Figure 6 diagnostics-14-02104-f006:**
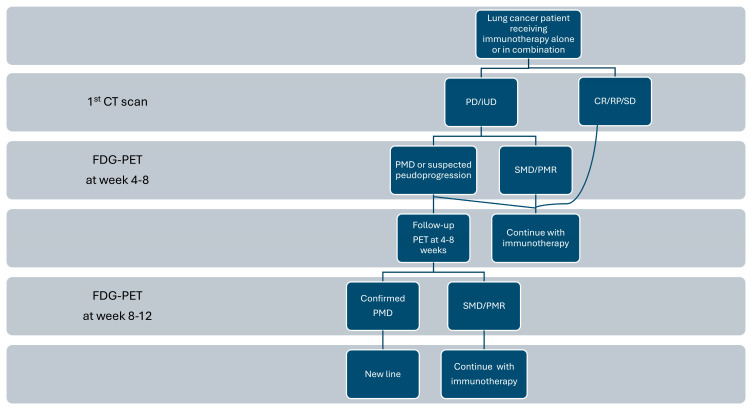
Algorithm for care pathway for patients with lung cancer receiving immunotherapy.

**Table 1 diagnostics-14-02104-t001:** Comparison of imRECIST with RECIST v1.1 and irRC [12]. CR: complete response; imRECIST: immune-modified RECIST; irRECIST: immune-related Response Evaluation Criteria In Solid Tumors; PD: progressive disease; RECIST: Response Evaluation Criteria In Solid Tumors; SLD: sum of longest diameters.

Criterion	RECIST v1.1	iRECIST	imRECIST
Tumor burden	Unidimensional Up to 5 target lesions/2 per organ	Bidimensional per WHO Up to 10 target lesions/5 per organ	Unidimensional, with other target lesion criteria (number and measurability) per RECIST v1.1
New lesions	Always represent PD	New lesions do not categorically define PD Measurable new lesions incorporated into the total tumor burden Nonmeasurable new lesions preclude CR	
Non-target lesions	Can contribute to defining CR or PD (unequivocal progression)	Non-target progression does not define PD, can only contribute to defining CR (complete disappearance required)	
PD	≥20% increase in the SLD (RECIST) and ≥5 mm increase compared with nadir, unequivocal progression in non-target lesions, and/or appearance of new lesions	Determined only on the basis of measurable disease	
		Negated by subsequent non-PD assessment ≥4 weeks from the date first documented (lack of confirmation)	
	Confirmation of PD not required	≥25% increase in the SLD compared with baseline/nadir	≥20% increase in SLD (RECIST) compared with baseline/nadir
		Best response may occur before confirmed PD	Best response may occur after any number of PD assessments

**Table 2 diagnostics-14-02104-t002:** Immune-related response criteria: [18F]FDG PET adapted from [14].

Criteria	EORTC	PERCIST 1.0	PECRIT	PERCIMT	iPERCIST	imPERCIST5
**Reference**	Young et al. [15]	Wahl et al. [16]	Cho et al. [17]	Anwar et al. [18]	Goldfarb et al. [19]	Ito et al. [20]
**Year**	1999	2009	2017	2018	2019	2019
**Tumor**	Solid tumors	Solid tumors	Melanoma	Melanoma	NSCLC	Melanoma
**Treatment**	Chemotherapy	Chemotherapy and targeted therapies	Immune checkpoints inhibitors (anti-PD1, anti-CTLA4)	Immune checkpoints inhibitors (anti-CTLA4)	Immune checkpoints inhibitors (anti-PD1)	Immune checkpoints inhibitors (anti-PD1)
**Modality**	FDG-PET	FDG-PET	CT and FDG-PET	CT and FDG-PET	FDG-PET	FDG-PET
**Delay for confirmation of progressive metabolic disease (PMD)**	Undetermined	Undetermined	3–4 weeks	3 months	2 months	3 months
**Target lesions**	Tumor lesion with the highest SUV uptake	Minimum tumor SUL 1.5× mean SUL liver, >5 target lesions/patient	RECIST 1.1 PERCIST 1.0	Size (metabolically active lesion) >1.0 or 1.5 cm, ≤5 target lesions/patient	Minimum tumor SUL 1.5× mean SUL liver, >5 target lesions/patient	Minimum tumor SUL 1.5× mean SUL liver, >5 target lesions/patient
**New lesions**	Progression	Progression	Progression	Metabolic progressive disease (PMD)	Immune unconfirmed metabolic progressive disease (iuPMD)	Need to be included in the SULpeak, PMD if >30% increase in SULpeak
**Complete metabolic response (CMR)**	Complete resolution of FDG uptake within the tumor volume so that it was indistinguishable from surrounding normal tissue	Disappearance of all metabolically active lesions	Disappearance of all lesions	Disappearance of all metabolically active lesions	Disappearance of any uptake in target lesion	Disappearance of all metabolically active lesions
**Partial metabolic response (PMR)**	A reduction of a minimum of 15–25% in tumor SUVs after one cycle of chemotherapy and >25% after more than one treatment cycle	Reduction in SULpeak in target lesions > 30% and absolute drop in SUL > 0.8 SUL units	≥30% decrease from baseline	Disappearance of some metabolically active lesions without any new lesion	Reduction in SULpeak in target lesions ≥30%	Reduction in SULpeak in target lesions by ≥30% and absolute drop in SUL by ≥0.8 SUL units
**Stable metabolic disease (SMD)**	An increase in SUVs <25% or a decrease <15% and no visible increase in extent of FDG tumor uptake (>20% in the longest dimension)	Neither PMD, PMR, nor CMR	Neither PD, PR, nor CR, evaluation of change in SULpeak of the hottest lesion: >15.5% (clinical benefit), ≤15.5% (no clinical benefit)	Neither PMD, PMR, nor CMR	Neither PMD, PMR, nor CMR	Neither PMD, PMR, nor CMR
**Progressive metabolic disease (PMD)**	An increase in SUVs >25% within the tumor region defined on the baseline scan, visible increase in the extent of FDG tumor uptake (>20% in the longest dimension) or the appearance of new FDG uptake in metastatic lesions	Increase in SULpeak of > 30% or the appearance of a new lesion	≥20% increase in the nadir of the sum of target lesions (>5 mm)	≥4 new lesions of <1 cm or ≥3 new lesions of >1 cm or ≥2 new lesions of >1.5 cm	≥30% increase in SULpeak or new metabolically active lesions: immune unconfirmed PMD (iuPMD)	>30% increase in SULpeak, with >0.8 SUL unit increase in tumor SULpeak
**Confirmed PMD** **(cPMD)**	n.a.	n.a.	n.a.	n.a.	PET at 4–8 weeks: confirmed PMD?	n.a.

## Data Availability

No new data were created or analyzed in this study. Data sharing is not applicable to this article.

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
