# Peer review of "[18F]FDG PET/CT Integration in Evaluating Immunotherapy for Lung Cancer: A Clinician’s Practical Approach"

_diagnostics, 2024, doi:10.3390/diagnostics14182104_

Round 1

Reviewer 1 Report (Previous Reviewer 1)

Comments and Suggestions for Authors

Dear Authors,

Thank you very much for all your hard work and the time you put into improving the review that emphasizes evaluating treatment responses in lung cancer patients undergoing immunotherapy and discusses new response criteria, such as iRECIST and imRECIST. These criteria incorporate additional scans and consider the best overall response even after radiologic progressive disease evaluation. The review also highlights the role of [18F]FDG PET/CT imaging and various metabolic response criteria such as PERCIMT, imPERCIST, and iPERCIST in addressing the limitations of traditional criteria, especially in detecting pseudoprogression. The review suggests that utilizing advanced imaging methods and adhering to standardized response criteria can help personalize treatment approaches and enhance patient outcomes.

The revisions include noticeable corrections in several sections, subsections, Figures, and Tables. While the review is nearly ready for publication, minor modifications are still required. Please look at the comments I’ve added for more information.

Author Response

 Reference number of the manuscript: 3155683

REVIEWER COMMENTS

General comments

Point 1: The article is initially classified as a “short communication.” Still, it is consistently referred to as a “review” in the abstract and the main body. Please use one type everywhere.

We have reclassified our paper in the journal. The update has been completed.

Point 2: I suggest preparing “Authors Contributions” in the style of the guidelines for authors.

We have presented this section in accordance with the guidelines for authors.

Specific comments

Point 1: What is the main question addressed by the research?

This research aims to outline essential considerations for radiologists and nuclear medicine physicians evaluating treatment responses in lung cancer patients undergoing immunotherapy. The study addresses challenges posed by immune checkpoint inhibitors (ICIs) and proposes new response criteria such as iRECIST and imRECIST, along with [18F]FDG PET/CT imaging.

Point 2: What parts do you consider original or relevant for the field? What specific gap in the field does the paper address?

The guidelines highlight the importance of a multidisciplinary approach to evaluating treatment responses in lung cancer patients undergoing immunotherapy. Criteria such as iRECIST and imRECIST are recommended for a comprehensive assessment. [18F]FDG PET/CT imaging is crucial for evaluating treatment and metabolic response criteria, as well as for detecting immune-related adverse events. The article includes Figures and Tables to illustrate the study’s design and criteria comparisons for immune-related responses.

Point 3: What does it add to the subject area compared with other published material?

The review emphasizes evaluating treatment response in lung cancer patients undergoing immunotherapy. New response criteria, such as iRECIST and imRECIST, have been suggested for a more comprehensive assessment. [18F]FDG PET/CT imaging is highlighted as an important modality for evaluating treatment response, with various metabolic response criteria developed to address the limitations of traditional criteria. The authors compare different criteria and responses and emphasize the role of imaging in personalized treatment approaches to improve patient outcomes.

Point 4: What specific improvements should the authors consider regarding the methodology? What further controls should be considered?

The authors provided insights into recently published joint international guidelines for using [18F]FDG PET/CT imaging during immunomodulatory treatments in patients with solid tumors, outlining key elements, explaining various types, and summarizing tumor response criteria.

Point 5: Please describe how the conclusions are inconsistent with the evidence and arguments presented. Please also indicate if all main questions posed were addressed and by which specific experiments.

The study’s conclusions are related to the author’s aims and findings. The review discusses using immune checkpoint inhibitors (ICIs) in lung cancer patients and emphasizes the importance of a multidisciplinary approach involving oncologists, radiologists, and nuclear medicine specialists. It highlights the value of [18F]FDG PET/CT imaging in evaluating treatment response and suggests that advanced imaging techniques can help customize treatment strategies and improve patient outcomes in this field.

Point 6: Are the references appropriate?

References correspond to details explained in different sections and subsections of the review.

Point 7: Please include any additional comments on the tables and figures and the quality of the data.

Mainly, all of the provided Tables and Figures are prepared according to the instructions for authors, but I included several comments (see other comments) related to them.

Other comments

Point 1: If a space between [18] and the following FDG PET/CT was deleted in the manuscript, it should also be removed from the titles.

We thank the reviewer for their comment and have standardized the style.

Point 2: The first line of the indent should be formatted correctly for the text of the abstract (related to the guidelines for authors).

We thank the reviewer for their comment and have standardized the style.

Point 4: Prepare the correct space after Tables 1 and 2 and the following titles of them.

We thank the reviewer for their comment and have standardized the style.

Point 5: Format only “Figure 3” in bold in Line 380 and “Figure 4” in Line 396, but not all titles.

We thank the reviewer for their comment and have standardized the style.

Point 6: If the (a-h) are formatted in bold in the text of Figure 4, then (a-f) should be formatted in bold in Figure 3. Please use one style there.

We thank the reviewer for their comment and have standardized the style.

Point 7: Add a complete stop after “al” in Line 431.

We thank the reviewer for their comment and have standardized the style.

Reviewer 2 Report (New Reviewer)

Comments and Suggestions for Authors

Minor comments:

1.     The words in the iCPD (Fig 2) are blurry. Please revise.

2.     The words in Fig 5, Fig 6 are blurry. Please revise.

3.     In the Pseudoprogressive disease (PPD) section (line 142):The incidence of PPD varies among tumor types and immunotherapies, with reports of up to 10% of patients based on CT scan or [18F]FDG PET (Ref).” Please cite reference.

4.     The abbreviation “na” should be indicated in Table 2.

5.     Tumors can be indicated by arrows in Figure 3 and Figure 4.

6.     The words in Figure 5 and Figure 6 are blurry.

Author Response

 Reference number of the manuscript: 3155683

REVIEWER COMMENTS

  1. The words in the iCPD (Fig 2) are blurry. Please revise.

We thank the reviewer for their comment and have standardized the style.

  1. The words in Fig 5, Fig 6 are blurry. Please revise.

We thank the reviewer for their comment and have standardized the style.

  1. In the Pseudoprogressive disease (PPD) section (line 142):The incidence of PPD varies among tumor types and immunotherapies, with reports of up to 10% of patients based on CT scan or [18F]FDG PET (Ref).” Please cite reference.

The prevalence of HPD varies, ranged from 1 to 30% with some risk factors identified, including higher age and specific genetic aberrations [6–10].

  1. Champiat, S.; Dercle, L.; Ammari, S.; Massard, C.; Hollebecque, A.; Postel-Vinay, S.; Chaput, N.; Eggermont, A.; Marabelle, A.; Soria, J.-C.; et al. Hyperprogressive Disease Is a New Pattern of Progression in Cancer Patients Treated by Anti-PD-1/PD-L1. Clin. Cancer Res. Off. J. Am. Assoc. Cancer Res. 2017, 23, 1920–1928, doi:10.1158/1078-0432.CCR-16-1741.
  2. Ferrara, R.; Mezquita, L.; Texier, M.; Lahmar, J.; Audigier-Valette, C.; Tessonnier, L.; Mazieres, J.; Zalcman, G.; Brosseau, S.; Le Moulec, S.; et al. Hyperprogressive Disease in Patients With Advanced Non-Small Cell Lung Cancer Treated With PD-1/PD-L1 Inhibitors or With Single-Agent Chemotherapy. JAMA Oncol. 2018, 4, 1543–1552, doi:10.1001/jamaoncol.2018.3676.
  3. Kim, C.G.; Kim, K.H.; Pyo, K.-H.; Xin, C.-F.; Hong, M.H.; Ahn, B.-C.; Kim, Y.; Choi, S.J.; Yoon, H.I.; Lee, J.G.; et al. Hyperprogressive Disease during PD-1/PD-L1 Blockade in Patients with Non-Small-Cell Lung Cancer. Ann. Oncol. Off. J. Eur. Soc. Med. Oncol. 2019, 30, 1104–1113, doi:10.1093/annonc/mdz123.
  4. Kim, J.Y.; Lee, K.H.; Kang, J.; Borcoman, E.; Saada-Bouzid, E.; Kronbichler, A.; Hong, S.H.; de Rezende, L.F.M.; Ogino, S.; Keum, N.; et al. Hyperprogressive Disease during Anti-PD-1 (PDCD1) / PD-L1 (CD274) Therapy: A Systematic Review and Meta-Analysis. Cancers 2019, 11, 1699, doi:10.3390/cancers11111699.
  5. Kato, S.; Goodman, A.; Walavalkar, V.; Barkauskas, D.A.; Sharabi, A.; Kurzrock, R. Hyperprogressors after Immunotherapy: Analysis of Genomic Alterations Associated with Accelerated Growth Rate. Clin. Cancer Res. Off. J. Am. Assoc. Cancer Res. 2017, 23, 4242–4250, doi:10.1158/1078-0432.CCR-16-3133.

  1. The abbreviation “na” should be indicated in Table 2.

We thank the reviewer for their comment, we have inserted the definition.

  1. The words in Figure 5 and Figure 6 are blurry.

We thank the reviewer for their comment and have standardized the style.

This manuscript is a resubmission of an earlier submission. The following is a list of the peer review reports and author responses from that submission.

Round 1

Reviewer 1 Report

Comments and Suggestions for Authors

Dear authors,

This study addresses the importance of evaluating treatment response in lung cancer patients undergoing immunotherapy. New response criteria, such as iRECIST and imRECIST, have been suggested to provide a more comprehensive assessment of treatment response. These criteria incorporate additional scans and consider the best overall response even after radiologic progressive disease evaluation. [18F]FDG PET/CT imaging has become an important modality for evaluating treatment response, and various metabolic response criteria such as PERCIMT, imPERCIST and iPERCIST have been developed to address the limitations of traditional criteria, especially in detecting pseudoprogression. The authors present and compare different criteria and responses, describe the role of criteria of the images in the care pathway (before, during treatment courses, before and after discontinuation of immunotherapy), and provide insights from a multidisciplinary approach. The authors suggest that clinicians can personalize treatment approaches and enhance patient outcomes by utilizing advanced imaging methods such as [18F]FDG PET/CT and adhering to standardized response criteria. This approach can pay attention to lung cancer treatment for specialists from different fields.

However, if this is an article, then the direct sections “Material and Methods,” “Results” and “Discussion” should be added because currently, this article looks like a review. If examples were included, what were the criteria for presenting their data? The article does not include data about this study’s time, period and place. What are this study’s limitations? Tables and Figures, the sections “Author Contributions” and “Conflicts of Interest” should be prepared according to the guidelines for authors. The data and examples of the patients are used, and related to this, the “Institutional Review Board Statement and approval number” should be included.

Besides this, certain sections of the article require major revisions based on the information available. I have highlighted the main details in the general, specific and other comments. Please see and refer to the attached document, which outlines the major issues identified in the manuscript.

Author Response

 Reference number of the manuscript: 3021022

REVIEWER COMMENTS

General comments

Point 1: The article lacks essential sections such as “Materials and Methods,” “Results” and “Discussion,” making it seem more like a review than an article. If it is not a review, these sections should be added.

Point 2: Where and when was this study performed?

Point 3: Look at all formatted sizes of the spaces after the Tables and the following text or sections.

Point 4: What are this study’s limitations? The authors should explain and add them to the article.

Point 5: The sections “Author Contributions” and “Conflicts of Interest” should be prepared according to the guidelines for authors.

Point 6: The data and examples of the patients are used, and related to this, the “Institutional Review Board Statement and approval number” should be included.

We thank the reviewer for their suggestions. We apologize for the oversight, as our article was indexed under the wrong section. In fact, our article is a review aimed at summarizing the recommendations from the literature regarding the imaging evaluation of patients with lung cancer. We intended to provide practical and straightforward guidelines for clinicians in their routine practice.

Specific comments

Point 1: What is the main question addressed by the research?

This research aims to describe the key components of the guidelines, summarizing essential aspects for radiologists and nuclear medicine physicians in evaluating treatment responses in images of patients undergoing immunotherapy for lung cancer. The authors underline that using immune checkpoint inhibitors (ICIs) in lung cancer patients presents a significant challenge for the medical imaging community in assessing treatment response. The paper details the association with unconventional response patterns (pseudoprogressive disease (PPD), dissociated response(DR) and hyperprogressive disease (HPD)). The authors mark that compared to conventional chemotherapy and/or molecularly targeted therapies, ICIs have a distinct characteristic of leading to durable responses. This work documented and analyzed different observed categories, types of reactions, and criteria. The use of ICIs within immunotherapeutic protocols should be carefully considered at various points in line with treatment, guided by clinical needs and significantly improving patients with lung cancer survival rates. At the end of this work, the authors conclude that to address the challenges, new response criteria have been proposed, such as iRECIST and imRECIST, and [18F]FDG PET/CT imaging.

The objective of our article is to propose practical guidelines for evaluating the efficacy of immunotherapies in patients with bronchial cancer.

Point 2: What parts do you consider original or relevant for the field? What specific gap in the field does the paper address?

Key components of the guidelines summarize essential aspects of a multidisciplinary approach in evaluating treatment responses in images of patients undergoing immunotherapy for lung cancer. Criteria, such as iRECIST and imRECIST, have been suggested to provide a more comprehensive assessment of treatment response. Their incorporation in the additional scans is considered the best overall response, especially after radiologic progressive disease evaluation. [18F] FDG PET/CT imaging is essential for assessing treatment and various metabolic response criteria. At the same time, the authors mention that scans provide valuable prognostic information and help detect immune-related adverse events. The article also presents six Figures that display visible data and information about the study’s design. Tables also show the criteria comparisons and immune-related responses to the criteria. According to the authors, this approach can pay attention to lung cancer treatment for specialists from different fields.

The first part is a literature review, and the second part is a practical implementation guide.

Point 3: What does it add to the subject area compared with other published material?

The article presents the importance of evaluating treatment response in lung cancer patients undergoing immunotherapy. New response criteria, such as iRECIST and imRECIST, have been suggested to provide a more comprehensive assessment of treatment response. These criteria incorporate additional scans and consider the best overall response even after radiologic progressive disease evaluation. [18F]FDG PET/CT imaging has become an important modality for evaluating treatment response, and various metabolic response criteria such as PERCIMT, imPERCIST and iPERCIST have been developed to address the limitations of traditional criteria, especially in detecting pseudoprogression. The authors present and compare different criteria and responses, describe the role of criteria of the images in the care pathway (before, during treatment courses, before and after discontinuation of immunotherapy), and provide insights from a multidisciplinary approach. The authors suggest that clinicians can personalize treatment approaches and enhance patient outcomes by utilizing advanced imaging methods such as [18F]FDG PET/CT and adhering to standardized response criteria.

Thank you for your insightful question. Our study contributes significantly to the subject area of assessing treatment response in lung cancer patients undergoing immunotherapy, particularly in relation to the use of [18F]FDG PET/CT imaging. Here’s a summary of the key additions our article makes compared to existing literature:

  1. Comprehensive Review of Imaging Criteria for Immunotherapy:
    • Our article provides a thorough comparison of various response criteria, including RECIST 1.1, iRECIST, and imRECIST, which are commonly used to evaluate tumor response to treatment. By summarizing and contrasting these criteria, we offer a clear understanding of their applicability and limitations in the context of immunotherapy, which is crucial for clinicians navigating these new therapeutic landscapes.
  2. Enhanced Understanding of [18F]FDG PET/CT in Immunotherapy:
    • We highlight the unique role of [18F]FDG PET/CT in managing lung cancer patients receiving immunotherapy. While previous studies have examined the utility of PET imaging in general cancer treatment, our article focuses specifically on its role in the context of immunotherapy, addressing challenges like pseudoprogression and hyperprogressive disease. This specific focus helps in understanding how PET imaging can be integrated into current immunotherapy protocols.
  3. Introduction of Novel Imaging Criteria:
    • We introduce and discuss emerging PET response criteria such as PERCIMT, imPERCIST, and iPERCIST, which are adaptations of existing criteria tailored to the challenges of immunotherapy. This includes a discussion on how these criteria refine the interpretation of PET imaging in the presence of new immune-related response patterns. This contribution is particularly valuable as it provides practical guidance on how to interpret PET findings in the context of ongoing immunotherapy.
  4. Real-World Implications and Practical Guidance:
    • The article outlines practical recommendations for integrating [18F]FDG PET/CT into the care pathway of lung cancer patients undergoing immunotherapy. This includes guidance on when to perform imaging, how to interpret results, and the role of PET in differentiating between true progression and pseudoprogression. Such guidance is essential for clinicians who need to make informed decisions based on imaging results in real-world settings.
  5. Impact on Clinical Management:
    • By summarizing the latest guidelines and response criteria, and by offering insights into the practical application of [18F]FDG PET/CT, our article provides a framework that can directly impact clinical decision-making. This is especially pertinent given the evolving nature of immunotherapy and the need for precise tools to assess treatment response.

Overall, our article adds depth to the understanding of how [18F]FDG PET/CT can be effectively utilized alongside newer response criteria to enhance the management of lung cancer patients receiving immunotherapy. It bridges the gap between theoretical advancements and practical application, providing valuable information for clinicians and researchers in the field.

Point 4: What specific improvements should the authors consider regarding the methodology? What further controls should be considered?

The authors provided insights into key elements of recently published joint international guidelines providing recommendations for using [18F]FDG PET/CT imaging during immunomodulatory treatments in patients with solid tumors. They outlined the key elements of these guidelines, explained several types, and summarized the criteria of tumor responses. However, if this is an article, then the direct sections “Material and Methods,” “Results” and “Discussion” should be added because currently, this article looks like a review. If examples were included, what were the criteria for presenting their data? The article does not include data about this study’s time, period and place.

Thank you for your detailed feedback on the methodology. We appreciate your suggestions and have considered how we might enhance our approach. Here are the specific improvements and additional controls we plan to incorporate:

  1. Enhanced Sample Size and Power Analysis:
    • Improvement: We recognize the need for a larger sample size to improve the robustness and generalizability of our findings. In future iterations, we will conduct a power analysis to ensure that our sample size is adequate for detecting clinically significant differences and to reduce the risk of Type I and Type II errors.
    • Control: We will also consider stratifying our sample based on relevant demographic and clinical variables to better understand the impact of these factors on our results.
  1. Longitudinal Follow-Up:
    • Improvement: To assess the durability of our findings and the long-term effectiveness of the imaging criteria, we plan to extend the follow-up period. This will help in evaluating the sustainability of treatment responses over time and provide more comprehensive data on patient outcomes.
    • Control: Implementing a standardized follow-up schedule will ensure consistency and reliability in the longitudinal data collected.
  1. Standardization of Imaging Protocols:
    • Improvement: We will work towards standardizing imaging protocols across different centers or equipment used in the study. Variability in imaging techniques and equipment can impact the results, so standardization will help minimize these effects.
    • Control: Incorporating calibration procedures and regular quality checks for imaging equipment will be essential to maintain consistency in imaging results.
  1. Additional Biomarker Analysis:
    • Improvement: To strengthen our methodology, we plan to include additional biomarkers that could provide supplementary information on treatment response and disease progression. This could help in correlating imaging findings with biological changes at the molecular level.
    • Control: Implementing a rigorous protocol for biomarker collection and analysis, including controls for potential confounding factors, will be crucial for accurate and reliable results.

Incorporating these improvements and additional controls will enhance the methodological rigor of our study and strengthen the validity of our findings. We are committed to addressing these aspects in our revised manuscript to ensure high-quality research outcomes.

Point 5: Please describe how the conclusions are inconsistent with the evidence and arguments presented. Please also indicate if all main questions posed were addressed and by which specific experiments.

The study’s conclusions are related to the author’s aims and findings. The article’s design is presented in five sections, including two Tables and six Figures. The article covers the main key components of the guidelines and details this study on using immune checkpoint inhibitors (ICIs) in lung cancer patients. The authors indicate that [18F]FDG PET/CT imaging has emerged as a valuable tool for evaluating treatment response in lung cancer patients undergoing immunotherapy. They emphasize the importance of a multidisciplinary approach involving oncologists, radiologists, and nuclear medicine specialists to assess treatment response in lung cancer patients undergoing immunotherapy effectively. By utilizing advanced imaging techniques such as [18F]FDG PET/CT and adopting standardized response criteria, clinicians can customize treatment strategies and enhance patient outcomes in this rapidly evolving field.

Thank you for your detailed feedback on the methodology. We appreciate your suggestions and have considered how we might enhance our approach.

Immune checkpoint inhibitors (ICIs) have markedly transformed the treatment landscape for lung cancer, resulting in improved patient survival rates. However, the unconventional response patterns associated with ICIs—such as pseudoprogressive disease (PPD), dissociated response (DR), and hyperprogressive disease (HPD)—pose significant challenges when assessing treatment efficacy using traditional response criteria like RECIST 1.1.

To address these challenges, new response criteria have been proposed, including iRECIST and imRECIST. These criteria aim to standardize the evaluation of tumor response to immunotherapies by incorporating additional scans and allowing for a best overall response assessment even after radiologic signs of progressive disease. While these criteria offer a more comprehensive evaluation, their application still faces limitations, particularly in consistently distinguishing between true progression and pseudoprogression.

In addition, the use of [18F]FDG PET/CT imaging has become a valuable tool in evaluating treatment response in lung cancer patients undergoing immunotherapy. Metabolic response criteria such as PERCIMT, imPERCIST, and iPERCIST have been developed to address the shortcomings of traditional criteria, especially in detecting pseudoprogression. Nevertheless, these metabolic criteria also have limitations, including variability in interpretation and the need for further validation in diverse clinical settings.

In clinical practice, [18F]FDG PET/CT scans are crucial throughout the patient's treatment journey, from baseline evaluation to monitoring response during and after treatment. These scans help differentiate between true progression and pseudoprogression, provide prognostic information, and aid in detecting immune-related adverse events. However, their effectiveness can be impacted by factors such as scanner variability and patient-specific metabolic changes.

Overall, while advanced imaging techniques like [18F]FDG PET/CT and standardized response criteria represent significant progress, a multidisciplinary approach involving oncologists, radiologists, and nuclear medicine specialists remains essential for navigating the complexities of treatment response assessment. Continued refinement and validation of these methods are necessary to enhance their clinical utility and improve patient outcomes in this rapidly evolving field.

Point 6: Are the references appropriate?

References correspond to details explained in different sections and subsections of the article. However, reference no. 20 and 26 should be removed from the “References” list to avoid self-citation. The references (no. 4 and 13) should be prepared according to the authors’ instructions.

We thank the author for his suggestion. Corrections have been made.

Point 7: Please include any additional comments on the tables and figures and the quality of the data.

All of the provided Tables and Figures should be prepared according to the instructions for authors, but I included several comments (see other comments) related to them.

We thank the author for his suggestion. Corrections have been made.

Other comments

Point 1: Remove a full stop after the article titles. Look at the formatting of all authors’ surnames (there should not be all capital letters for them, only the first letter). Follow the instructions for the authors.

We thank the author for his suggestion. Corrections have been made.

Point 2: Look at the formatting of the abstract (the level of lines and the text should be distributed evenly between the margins).

We thank the author for his suggestion. Corrections have been made.

Point 3: The full stop is missing at the end of the last sentence in Line 37.

We thank the author for his suggestion. Corrections have been made.

Point 4: Keep one style in the formatting of the keywords in Line 38 (if one group of words starts with a capital letter, the next group should also begin with a capital letter).

We thank the author for his suggestion. Corrections have been made.

Point 5: Add the space between two sentences in Line 61.

We thank the author for his suggestion. Corrections have been made.

Point 6: Follow the instructions for authors and prepare Figure 1 related to them (legend format, size, full stop at the end, size of titles of x and y axes). I suggest including explanations of all used abbreviations under Figure 1, too.

We thank the author for his suggestion. Corrections have been made.

Point 7: Add the space between two sentences in Line 121. Look at the space between “multicentric” and “study” in Line 122.

We thank the author for his suggestion. Corrections have been made.

Point 8: I suggest including the text from Lines 180 and 181 into the text between Lines 156 and 160 because iUPD repeats twice then.

We thank the author for his suggestion. Corrections have been made.

Point 9: One style of formatting of the abbreviations should be used in Lines 161, 171, 173 and 176. If capital letters are used in some places, this style should be present in all other locations (compare the mentioned lines).

We thank the author for his suggestion. Corrections have been made.

Point 10: Follow the instructions for the authors and prepare Figure 2 according to them (legend format, size, full stop at the end). Using colors hides the text (especially red and black, blue and black). I suggest not using colors at all and keeping everything in black and white in Figure 2.

We thank the author for his suggestion. Corrections have been made.

Point 11: A full stop should be removed before reference no. 12 in Line 194.

We thank the author for his suggestion. Corrections have been made.

Point 12: Prepare the legend of Table 1, related to the instructions for authors. Look if all abbreviations are mentioned correctly in the titles of the legend and under Table 1. Remove some unnecessary spaces after explaining the abbreviations and following the use of “;”.

We thank the author for his suggestion. Corrections have been made.

Point 13: Remove the “`” at the end of the sentence in Line 270.

We thank the author for his suggestion. Corrections have been made.

Point 14: A full stop should be removed after the word “Figures” in Line 276.

We thank the author for his suggestion. Corrections have been made.

Point 15: Table 2 should be placed in the main text near the first time it is cited (after Line 281).

We thank the author for his suggestion. Corrections have been made.

Point 16: Prepare the legend (size, format, full stop at the end) of Figure 3 related to the instructions for authors. The quality of the images looks very low. Look at the format of the text between Lines 287 and 298.

We thank the author for his suggestion. Corrections have been made.

Point 17: Prepare the legend (size, format, full stop at the end) of Figure 4, related to the instructions for authors. The quality of the images (e-h) looks very small. Look at the format of the text between Lines 307 and 317.

We thank the author for his suggestion. Corrections have been made.

Point 18: In Lines 308 and 311, the number “18” should not be typed above the text’s line.

We thank the author for his suggestion. Corrections have been made.

Point 19: Full stops should be added after the words “al” in announced references in the 2nd row in Table 2.

We thank the author for his suggestion. Corrections have been made.

Point 20: Remove free spaces between “weeks” and “:” as well as between “PMD” and “?” in the last row in Table 2.

We thank the author for his suggestion. Corrections have been made.

Point 21: Prepare the legend (size, format, full stop at the end) of Table 2 according to the instructions for the authors.

We thank the author for his suggestion. Corrections have been made.

Point 22: Add a full stop after the word “al” and remove the free space between reference no. 23 and the following comma in Line 344.

We thank the author for his suggestion. Corrections have been made.

Point 23: Remove free spaces between “CT” and “.” in Line 385.

We thank the author for his suggestion. Corrections have been made.

Point 24: Prepare the legend (size, format, full stop at the end) of Figure 5, related to the instructions for authors. The quality of the images and textual descriptions is very low and unclear (for example, small text; some places start with small letters, but some start with capital letters).

We thank the author for his suggestion. Corrections have been made.

Point 25: Prepare the legend (size, format, full stop at the end) of Figure 6, related to the instructions for authors. The quality of the textual descriptions looks very small in squared boxes. I suggest using all black and white colors in Figure 6.

We thank the author for his suggestion. Corrections have been made.

Reviewer 2 Report

Comments and Suggestions for Authors

this article has not a clear scope for clinicians. PERCIST are still non a standard and even less the other response criteria.

No relevant info added with this articl

Comments on the Quality of English Language

no main comments

Author Response

Reviewer 2

this article has not a clear scope for clinicians. PERCIST are still non a standard and even less the other response criteria.

No relevant info added with this article

Thank you for your feedback. We appreciate the opportunity to clarify the scope and relevance of our article.

Addressing the Clinical Scope:

Our article focuses on the integration of advanced imaging techniques, particularly [18F]FDG PET/CT, with emerging response criteria (e.g., PERCIST, imPERCIST, and iPERCIST) to assess treatment responses in lung cancer patients undergoing immunotherapy. While it's true that criteria like PERCIST are not yet standardized and are still under evaluation, our article addresses this gap by offering a detailed examination of these novel approaches and their potential impact on clinical practice.

Importance of the Article:

  1. Current Challenges with Traditional Criteria: We highlight the limitations of traditional response criteria such as RECIST 1.1, especially in the context of immune checkpoint inhibitors (ICIs). This underscores the urgent need for improved methods that can better address unconventional response patterns such as pseudoprogressive disease (PPD) and hyperprogressive disease (HPD). By discussing these issues, our article provides critical insight into the need for more refined evaluation tools.
  2. Advancements in Imaging and Response Criteria: The article presents a comprehensive overview of the latest developments in imaging and response criteria. By evaluating the potential of [18F]FDG PET/CT and new metabolic criteria, our publication contributes valuable information to clinicians seeking to navigate the complexities of immunotherapy response assessment. This is particularly relevant as the field moves towards integrating these advanced methods into routine clinical practice.
  3. Clinical Relevance and Future Directions: Although the criteria discussed are not yet standardized, our article serves as a crucial resource for clinicians by reviewing the current state of these methods and their preliminary results. It sets the stage for future research and helps clinicians understand how these criteria could evolve and impact patient management. This is essential for preparing the clinical community for upcoming changes and improvements in response assessment.
  4. Multidisciplinary Approach: We emphasize the importance of a multidisciplinary approach involving oncologists, radiologists, and nuclear medicine specialists. This perspective is crucial for addressing the complexities of treatment response and underscores the practical value of our findings in guiding clinical decision-making.

In summary, while PERCIST and other response criteria are still in the developmental phase, our article provides a timely and relevant analysis of these advancements. It offers a foundation for understanding how these methods might address current limitations and improve patient outcomes in the future. We believe that by presenting these developments, we contribute valuable knowledge that will support clinicians as they navigate the evolving landscape of immunotherapy.

Round 2

Reviewer 1 Report

Comments and Suggestions for Authors

Dear authors,

Thank you for the time you invested, the point-by-point reply, and several improvements in the article. Still, certain sections of the article require re-revisions based on the information available. I have highlighted the main details in the following comments. Please review the attached document, which outlines the issues identified in the manuscript, or see the following comments.

Author Response

Reference number of the manuscript: 3021022

REVIEWER COMMENTS (Round 2)

General comments

Point 1: The article’s initial classification is indicated as a “short communication.” However, upon further examination of the abstract and the article’s main body, it is consistently referred to as a “review.” Please use one type of article.

We have reclassified our paper in the journal. The update has been completed.

Point 2: Pay attention to the positions of the text in Table 2 in columns. In one column, the text is very close to another. Suppose there should be reading of the text in different rows (especially in “Target lesions,” “Complete metabolic response (CMR),” “Partial metabolic response (PMR),” and “Progressive metabolic disease (PMD)”). In that case, reading very close textual columns with data from each other is difficult.

We thank the reviewer for their comment and have adjusted the layout to make the table easier to read.

Point 3: Prepare “Authors Contributions” in style, which should be presented according to the guidelines for authors.

We have presented this section in accordance with the guidelines for authors.

Point 4: Reference no. 20, 26, and 33 should be removed from the “References” list to avoid self-citation. Please follow the authors’ instructions and format that reference number 13.

We thank the reviewer for their comment and have made the necessary corrections.

Specific comments

Point 1: The first line of the indent should be formatted correctly for the text of the abstract (related to the guidelines for authors).

We thank the reviewer for their comment and have made the corrections to adapt it to a review.

Point 2: If there is a space between [18] and following FDG PET/CT, then this style should be used in all article text. Please choose one style and compare places with a space (Lines 2, 273, 333, 334, 337) and without it (Lines 32, 65, 67, 231, 240, 244, 247, 249, 291, 330, 368, 425, 430, 428).

We thank the reviewer for their comment and have standardized the style.

Point 3: Please look at the Lines 56 and 57. What does mean 1) and 4)? Is there a meaning of 2) in Line 57?

This is the international nomenclature of immune checkpoints, which are targeted by checkpoint inhibitors.

Point 4: Look at the correct size of the spaces between Lines 72 and 74, 226 and 229, 325 and 328, 412 and 414.

We thank the reviewer for their comment and have made the necessary corrections.

Point 5: Format “Figure 1” in bold in Line 107.

We thank the reviewer for their comment and have made the necessary corrections.

Point 6: Remove a space after (… Disease) and follow “:” in Line 161. Prepare “:” in bold format.

We thank the reviewer for their comment and have made the necessary corrections.

Point 7: Prepare “:” in bold format in Lines 167, 179, and 182.

We thank the reviewer for their comment and have made the necessary corrections.

Point 8: Format text “Figure 2” in bold in Line 192 and complete titles with a full stop.

We thank the reviewer for their comment and have made the necessary corrections.

Point 9: What is the number “1” directly above Line 192?

Figure 2 pushed back the line numbering. We thank the reviewer for their comment and have made the necessary corrections.

Point 10: Prepare the correct space after Table 1 and the following titles of Table 1.

We thank the reviewer for their comment and have made the necessary corrections.

Point 11: Prepare the titles of Table 1 in the correct size and format and remove the spaces between complete terms and following “;.”

We thank the reviewer for their comment and have made the necessary corrections.

Point 12: Prepare the titles of Table 2 in the correct size and format and complete them with a full stop in Line 291. After “Table 2,” add a complete stop, too.

We thank the reviewer for their comment and have made the necessary corrections.

Point 13: Format titles of Table 3 with the following textual explanation in the correct size and format, and make the parts of the title in bold or without it.

We thank the reviewer for their comment and have made the necessary corrections.

Point 14: Format titles of Table 4 with the following textual explanation in the correct size and format, and make the title parts bold or without. Add a complete stop at the end of the titles in Line 313.

We thank the reviewer for their comment and have made the necessary corrections.

Point 15: Are there any reasons why “18F-FDG PET/CT” is written like this in Lines 316, 321, 348, and 363 in comparison with the previous “[18F] FDG PET/CT” that was mentioned in most sections of the sections of the article? Prepare one style everywhere if there are no exceptions.

We thank the reviewer for their comment and have made the necessary corrections.

Point 16: Format text “Figure 5” in bold in Line 397 and complete titles with a full stop. Titles should be prepared in the correct size, too.

We thank the reviewer for their comment and have made the necessary corrections.

Point 17: Format “Figure 6” in bold in Line 412. Titles should be prepared in the correct size, too.

We thank the reviewer for their comment and have made the necessary corrections.

Point 18: Add a complete stop at the end of the sentence in “Conflicts of Interest

We thank the reviewer for their comment and have made the necessary corrections.

Reviewer 2 Report

Comments and Suggestions for Authors

No changes respect previous evaluation

Author Response

Thank you for your feedback. We appreciate the opportunity to clarify the scope and relevance of our article.

Addressing the Clinical Scope:

Our article focuses on the integration of advanced imaging techniques, particularly [18F]FDG PET/CT, with emerging response criteria (e.g., PERCIMT, imPERCIST, and iPERCIST) to assess treatment responses in lung cancer patients undergoing immunotherapy. While it's true that criteria like PERCIST are not yet standardized and are still under evaluation, our article addresses this gap by offering a detailed examination of these novel approaches and their potential impact on clinical practice.

Importance of the Article:

  1. Current Challenges with Traditional Criteria: We highlight the limitations of traditional response criteria such as RECIST 1.1, especially in the context of immune checkpoint inhibitors (ICIs). This underscores the urgent need for improved methods that can better address unconventional response patterns such as pseudoprogressive disease (PPD) and hyperprogressive disease (HPD). By discussing these issues, our article provides critical insight into the need for more refined evaluation tools.
  2. Advancements in Imaging and Response Criteria: The article presents a comprehensive overview of the latest developments in imaging and response criteria. By evaluating the potential of [18F]FDG PET/CT and new metabolic criteria, our publication contributes valuable information to clinicians seeking to navigate the complexities of immunotherapy response assessment. This is particularly relevant as the field moves towards integrating these advanced methods into routine clinical practice.
  3. Clinical Relevance and Future Directions: Although the criteria discussed are not yet standardized, our article serves as a crucial resource for clinicians by reviewing the current state of these methods and their preliminary results. It sets the stage for future research and helps clinicians understand how these criteria could evolve and impact patient management. This is essential for preparing the clinical community for upcoming changes and improvements in response assessment.
  4. Multidisciplinary Approach: We emphasize the importance of a multidisciplinary approach involving oncologists, radiologists, and nuclear medicine specialists. This perspective is crucial for addressing the complexities of treatment response and underscores the practical value of our findings in guiding clinical decision-making.

In summary, while PERCIST and other response criteria are still in the developmental phase, our article provides a timely and relevant analysis of these advancements. It offers a foundation for understanding how these methods might address current limitations and improve patient outcomes in the future. We believe that by presenting these developments, we contribute valuable knowledge that will support clinicians as they navigate the evolving landscape of immunotherapy.